# Nrf2-Mediated Dichotomy in the Vascular System: Mechanistic and Therapeutic Perspective

**DOI:** 10.3390/cells11193042

**Published:** 2022-09-28

**Authors:** Weiwei Wu, Andrew Hendrix, Sharad Nair, Taixing Cui

**Affiliations:** 1School of Basic Medicine, Qingdao Medical College, Qingdao University, Qingdao 266071, China; 2Department of Cell Biology and Anatomy, University of South Carolina School of Medicine, Columbia, SC 29209, USA; 3Columbia VA Health System, Wm. Jennings Bryan Dorn VA Medical Center, Columbia, SC 29209, USA

**Keywords:** Nrf2, cardiovascular disease, endothelial cells, vascular smooth muscle cells

## Abstract

Nuclear factor-erythroid 2-related factor 2 (Nrf2), a transcription factor, controls the expression of more than 1000 genes that can be clustered into different categories with distinct functions ranging from redox balance and metabolism to protein quality control in the cell. The biological consequence of Nrf2 activation can be either protective or detrimental in a context-dependent manner. In the cardiovascular system, most studies have focused on the protective properties of Nrf2, mainly as a key transcription factor of antioxidant defense. However, emerging evidence revealed an unexpected role of Nrf2 in mediating cardiovascular maladaptive remodeling and dysfunction in certain disease settings. Herein we review the role of Nrf2 in cardiovascular diseases with a focus on vascular disease. We discuss the negative effect of Nrf2 on the vasculature as well as the potential underlying mechanisms. We also discuss the clinical relevance of targeting Nrf2 pathways for the treatment of cardiovascular and other diseases.

## 1. Introduction

According to the statistical data of World Health Organization, cardiovascular disease (CVD) is the underlying cause of death for approximately 17.9 million global deaths each year. CVDs, such as hypertension, atherosclerosis, aneurysm, myocardial hypertrophy, and coronary heart disease, among others are the leading cause of death globally. Despite the advance in cardiovascular research, it is still far from a comprehensive understanding of CVD. Effective therapies for CVD are limited.

The causes of heart and vascular diseases may vary; however, most share an important pathological mechanism: oxidative stress [1,2,3,4], which is resulted from an imbalance between the production and detoxification of reactive oxygen species (ROS) [5]. Uncontrolled ROS cause arrhythmias and induce cardiac hypertrophy, apoptosis, and necrosis to promote cardiac pathological remodeling and dysfunction leading to heart failure [6,7]. Similarly, oxidative stress causes vascular damage and disease [8]. It is worthy to note that under physiological conditions, ROS also act as signaling molecules to regulate a variety of cellular functions in the cardiovascular system, such as endothelial cell (EC) and vascular smooth muscle cell (VSMC) proliferation, migration, and apoptosis, as well as angiogenesis, vascular tone, host defense, and genomic stability [9]. Anti-oxidative stress appears to be a therapeutic approach for the prevention and treatment of cardiovascular diseases; however, how to precisely eliminate oxidative stress without affecting physiological redox balance remains a challenge [1].

The nuclear factor-erythroid 2-related factor 2 (Nrf2), a multifunctional transcription factor, controls the basal and induced expression of more than 1000 genes that can be divided into several groups with different functions ranging from antioxidant defense and detoxification to metabolism and protein quality control [10]. Previously, Nrf2 has been identified as a principal transcription factor for antioxidant defense, thereby providing cardiovascular protection. However, Nrf2 can also act as a mediator of cardiovascular maladaptive remodeling and dysfunction, particularly in chronic conditions, such as sustained pressure overload and diabetes [11,12,13]. We have recently outlined and discussed the emerging dark side of Nrf2 in the heart and the potential mechanisms of Nrf2-mediated myocardial damage and dysfunction [14]. On the other hand, whilst most of the reported functions of Nrf2 in the vasculature appear to be positive, a handful of evidence reveal an adverse role of Nrf2 in promoting vascular disease. Nevertheless, the enthusiasm for developing Nrf2 activators to treat oxidative stress-associated diseases including CVD is still high. Therefore, we describe the Nrf2 signaling pathway and sort out the bidirectional role of Nrf2 signaling pathway in CVD, especially in vascular diseases. As a result, the outcome underscores the ‘dark’ side of Nrf2 while unveiling clues for investigating the nature of the Nrf2-mediated dichotomy in the cardiovascular system, thereby providing reliable and specific targets for clinical treatment.

## 2. Nrf2 Signaling

### 2.1. Basic Functions

Nrf2 was originally isolated as a homolog of the hematopoietic transcription factor NF-E2 (nuclear factor-erythroid factor 2) p45, as such Nrf2 was initially known as NF-E2-related factor 2 [15].

Both Nrf2 and NF-E2 belong to the Cap “n” Collar (CNC) family which contains basic a leucine zipper (bZip) structure. To date, six members of the family have been identified, namely: NF-E2, Nrf1-3, Bach 1 [broad-complex, tram-track, bric-a-brac (BTB), CNC homolog 1], and Bach 2.

The differences between these CNC family members have been extensively reviewed [14,16,17]. In general, NF-E2 and Nrf1-3 usually act as transcriptional activators, while Bach 1 and 2 function as transcriptional repressors. The differentiation of committed erythroid progenitor cells involves the transcription factor NF-E2 [18]. Megakaryocyte development and maturation are also governed by the activity of NF-E2 [19,20]. Nrf1-3 proteins are widely expressed in the body. Nrf1 helps regulate the basal expression levels of certain cytoprotective enzyme genes but does not affect their inducible expression [21]. Whilst Nrf3 plays a minor role in the regulation of Phase II enzyme genes [22,23], it has a crucial role in SMC differentiation from stem cells [24,25]. Nrf2 is widely expressed in oxygen-consuming organs, such as skeletal muscle, heart, blood vessels, liver, kidney, brain, lung, skin, and digestive tract [26]. Nrf2 orchestrates the basal and induced expression of genes by combining to a cis-acting enhancer with a core nucleotide sequence of 5′-RTGACNNNGC-3′23, which is referred to as the antioxidant response element (ARE). To date, over 1000 Nrf2 target genes have been identified. These genes can be clustered into several functional groups including antioxidant, detoxification, metabolism, transcription factors, proteasomal and autophagic degradation, cell proliferation, and cell survival [27,28,29]. Thus, this allows Nrf2 to exert not only antioxidant defense but also other multifunctional activities.

### 2.2. Protein Structural Domain of Nrf2

Nrf2 is the most potent member of the CNC transcription factor family. It contains 605 amino acids and is divided into 7 highly conserved erythroid-derived CNC homology (ECH) protein domains, ranging from Neh1 to Neh7, based on their functions [27,29,30]. From the N-terminal to the C-terminal, the Nrf2 protein comprises Neh2, Neh4-Neh5, Neh7, Neh6, Neh1, and Neh3 (Figure 1A). Neh2 contains a low-affinity DLG-binding motif and a high-affinity ETGE-binding motif [31], which bind to the double glycine repeat (DGR) domain of Kelch-like ECH associating protein 1 (Keap1) (Figure 1B), a major negative regulator of Nrf2 [32]. Neh2 acts as an intermediate in the formation of heterodimer Nrf2 and Keap1. Keap1 mediates the ubiquitination and degradation of Nrf2 and is a natural inhibitor of Nrf2 in the cytoplasm. Neh4 and Neh5 are independent of each other and jointly regulate the trans-activation of cytoprotective genes. Neh4 and Neh5 both independently and collaboratively bind to CBP [CREB (cAMP Responsive Element Binding protein) Binding Protein], thus attaining maximum activation of antioxidant gene expression [33]. Of note, abnormal activation of Nrf2 in cancer cells can confer cytoprotection to them, thereby reducing the therapeutic effect of chemotherapy and radiotherapy. The increased cytoprotection is resulted from the increased interaction between Nrf2 and CBP, that is mediated by the Neh4 and Neh5 domains of Nrf2. Disruption of this interplay through small-molecule therapeutics negate the aberrant activation of Nrf2 in cancer cells [34]. The Neh4 and Neh5 domains of Nrf2 also directly interact with the C/H3-containing C terminus of p300. Acetylation of Nrf2 by p300 during the antioxidant reaction enhances promoter-specific DNA binding of Nrf2 [35]. Nrf2 activation can be modulated and directly controlled through interactions between the Neh4 and Neh5 domains and the RAC3 protein [36]. The Neh7 domain is a newly discovered domain located between the Neh5 and Neh6 domains. The interplay between the Neh7 domain and 9-cis retinoic acid (RA) can suppress Nrf2 activity [37,38]. Like the Neh2 domain, the Neh6 domain contains two important motifs, the DSGIS and DSAPGS motifs. Unlike the Neh2 domain, these two motifs of the Neh6 domain mediate the keap1 independent degradation of Nrf2 in stressed cells. These two important motifs of Neh6 recruit the dimeric β–transducin repeat-containing protein (β-TrCP) [39,40]. Glycogen synthase kinase-3 (GSK-3) phosphorylates a particular serine residue in the Neh6 structural domain of Nrf2, which forms a degradable structural domain. This degradable structural domain is then recognized by the ubiquitin ligase adaptor β-TrCP for Nrf2 ubiquitination and subsequent proteasomal degradation by the Cullin1/ring-box 1 (RBX1) complex [41,42]. The Neh1 structural domain forms a heterodimer with the small muscle neurofibrosarcoma (sMaf) protein through the CNC-bZIP region and binds to AREs on target genes to regulate transcription [43,44]. It contains the functional nuclear localization signal (NLS) and nuclear export signal (NES) sequences [45,46]. The C-terminal Neh3 domain of Nrf2 is also important for its activity. It may function as a transcriptional activation structural domain and may be engaged in the interaction of transcriptional apparatus components that affect its transcriptional activity [47].

### 2.3. Nrf2 Transcription-Related

The aryl hydrocarbon receptor (AHR) directly binds to the xenobiotic response element (XRE)-like element located at the Nrf2 promoter and modulates Nrf2 gene transcription [48,49]. Notch signaling directly triggers the Nrf2 stress adaptation response pathway by recruiting the Notch intracellular structural domain (NICD) transcriptome to the conserved Rbpjκ site of the Nrf2 promoter [50]. TGFβ1-induced epithelial-mesenchymal transition occurs through the transcription of Notch4 via Nrf2-dependent promoter activation [51]. In addition, hypermethylation or single nucleotide polymorphism (SNP) in the Nrf2 promoter region results in reduced Nrf2 expression [52].

### 2.4. Ubiquitinated Degradation of Nrf2

The intracellular half-life of Nrf2 is less than 20 min. However, Keap1 plays a major role in the regulation of protein stability and transcriptional activity of Nrf2 [16,53]. Alternatively, GSK-3 and β-TrCP-mediated proteasomal degradation of Nrf2 is not associated with Keap1. A recent review provides a comprehensive discussion of these issues [14]. Thus, a concise summary is made as follows. The Keap1-Nrf2 pathway: Keap1 contains five domains including NTR (the N-terminal region), BTB, IVR (the intervening region), DGR or Kelch, and CTR (the C-terminal region) (Figure 1B). BTB and Kelch are two major domains. BTB is responsible for the interaction of the Keap1 homodimer with the Cul3-Rbx1-E3 ligase complex. Meanwhile, Kelch is combined with the DLG and ETGE motifs in Neh2 of Nrf2. Nrf2 binds to one of the Keap1 homodimers via ETGE motif and DLG motif, forming firstly an “open” conformation and then a “closed” conformation, with subsequent ubiquitination and degradation of Nrf2 by Clu3-Rbx1-E3 ligase and subsequent release of Keap1. Under normal conditions, Keap1 continuously targets Nrf2 for degradation (Figure 1C). However, under stressful conditions, some oxidizing or electrophilic molecules react with the cysteine residues of Keap1, leading to conformational changes in Keap1, thereby reducing Keap1-mediated ubiquitination and degradation of Nrf2 (Figure 1D). The GSK-3β-Nrf2 pathway: Under non-stressed conditions, AKT phosphorylates the N-terminal pseudo-substrate structural domain of GSK-3, thereby leaving GSK-3 in an inactive state. However, once AKT is inhibited or inactivated, GSK-3 is activated and then phosphorylates the Neh6 of Nrf2, which in turn recruits β-TrCP for β-TrCP-mediated the proteasomal degradation of the Nrf2. Typically, the Keap1-Nrf2 pathway happens in the cytosol, whereas the GSK-3β-Nrf2 pathway occurs in the nucleus. The specific mechanisms and relative importance of Keap1 and GSK-3-TrCP in regulating Nrf2 degradation remain poorly understood.

## 3. Nrf2 and Vascular Physiology and Pathology

### 3.1. Nrf2 and Endothelial Cells

ECs are one of the major cell types in the vascular wall. ECs sprouting from existing vessels forms new blood vessels, named angiogenesis, which is a critical event in embryonic development and multiple disease processes. Global and EC-specific knockout of Nrf2 in mice demonstrates a critical and cell-autonomous role of Nrf2 in ECs for promoting angiogenesis [54]. As ECs are constantly exposed to hemodynamic forces such as shear stress—the frictional force from the blood flow acting on the surface of the vascular lumen [55,56], they sense the shear stress to govern both short-term vascular tone and long-term vascular remodeling to adjust vessel diameters to tissue demand. In mechanobiology, disturbed flow and unidirectional laminar flow are two important types of shear stress that exert differential effects on the functions of ECs through mechanosensitive transcription factor-dependent gene expression [57]. Nrf2 is an important mechanosensitive transcription factor that is upregulated after exposure to unidirectional laminar flow [57,58,59,60]. In the sections below, we will review the specific role of Nrf2 in endothelial function.

In ECs, the laminar flow may use specific mechanoreceptor(s) to activate the Nrf2/ARE pathway. Nrf2 is activated by laminar shear stress through the PI3K-AKT signaling pathway [61,62]. However, when the shear stress on the vessel wall is disturbed, the oscillatory flow leads to reduced NO production and increased superoxide release [63]. This reduces Nrf2-mediated activation of ARE-linked cytoprotective genes [64]. In contrast, Nrf2 deletion or knockdown suppresses laminar shear stress-induced cytoprotective effects [58]. In addition to the known role of Nrf2 as a mechanosensitive transcription factor and oxidative stress inhibitor, Nrf2 is also involved in the inflammatory response of ECs. Nrf2 prevents ECs at atherosclerosis-protected sites from exhibiting a proinflammatory state by inhibiting the p38-VCAM-1 signaling pathway [65]. Protein kinase C (PKC)1 activates the Nrf2 signaling pathway and induces heme oxygenase (HO)-1 in the vascular endothelium to enhance resistance to inflammation to maintain vascular homeostasis [66]. Curcumin treatment of cultured human arterial ECs also induces HO-1 expression through activation of Nrf2 and inhibits acute vascular inflammation [67]. Angiotensin II (Ang II) is closely involved in endothelial dysfunction by induction of apoptosis and oxidative stress [68], and the induced endothelial dysfunction is a high-risk factor for CVD [69]. Sirtuin (SIRT) 6 protects vascular ECs from Ang II-induced apoptosis and oxidative stress via activating Nrf2/ARE signaling [70]. The SIRT6/Nrf2/ARE signaling pathway is a key regulator of redox homeostasis in vascular ECs. Ang II decreases antioxidant potential and increases oxidative stress and inflammatory responses, thereby causing damage to human umbilical cord vascular endothelial cells (HUVECs) by blunting the Nrf2/ERK1/2/Nox2 (NADPH oxidase 2) system.

Nrf2 has been extensively studied in cigarette smoke-induced emphysema and chronic obstructive pulmonary disease (COPD). Compared with wild-type mice, Nrf2-deficient mice had earlier-onset and more extensive cigarette smoke-induced emphysema associated with more pronounced oxidative stress [71,72]. Whilst Nrf2-mediated antioxidant defense has been well demonstrated in lung epithelial cells [72,73], the activation of a mitogen-activated protein kinase (MPAK)/Nrf2/HO-1 pathway was proposed to underlie ginkgo biloba extract-induced protection against cigarette smoke-induced oxidative stress and cell death in human pulmonary artery ECs [74]. These studies indicate that Nrf2-operated antioxidant defense is a common mechanism against smoking-induced damage in various types of lung cells including ECs. In addition, such a notion was extrapolated into smoking-induced atherosclerosis and cerebrovascular injury [75,76]. Knockdown of Nrf2 markedly enhanced cigarette smoke-induced ROS production and NLRP3 (NLR family pyrin domain containing 3) inflammasome activation in human aortic ECs [75]. Metformin could activate Nrf2 pathway and protect against cigarette smoke-induced damage to brain ECs and loss of blood–brain barrier [76]. Given that Nrf2 knockout increased susceptibility to brain edema and blood–brain barrier breakdown in a mouse model of subarachnoid hemorrhage [77], it is likely that Nrf2 is a critical mediator of metformin-induced brain protection. Taken together, these findings underscore a beneficial effect of Nrf2 activation in ECs.

Recently, several studies have reported that natural compounds can inhibit Ang II-induced ECs injury through the Nrf2 signaling pathway. Briefly, celastrol, known as tretinoin, is a functional ingredient of Trypterygiun wilfordii Hook F that has powerful antioxidant and anti-inflammatory properties. Celastrol effectively attenuates Ang II-mediated ECs injury by activating the Nrf2/ERK1/2/Nox2 pathway [78]. Epigallocatechin-3-gallate (EGCG) is the main chemical compound of green tea, which has anti-inflammatory, antioxidant, and anti-angiogenic effects. EGCG ameliorates Ang II-induced oxidative stress and apoptosis in HUVECs through activation of the Nrf2/Caspase-3 signaling pathway [79]. Schisandrin C (SchC) is a dibenzocyclooctadiene derivative of Schisandra Chinensis with antioxidant properties. SchC targets Keap1 to attenuate oxidative stress by activating the Nrf2 pathway in the Ang II-stimulated vascular endothelium [80]. Osthole is the main active component of the herbal fruit with anti-inflammatory and antioxidant activities. Osthole significantly attenuates Ang II-induced apoptosis in rat aorta endothelial cells (RAECs) by reducing inflammation and oxidative stress by targeting the NF-κB pathway and Keap-1/Nrf2 pathway [81]. In addition, tilapia by-product oligopeptides protect against Ang II-induced endothelial injury in HUVECs in vitro via the activation Nrf2/NF-κB signal pathway [82]. Synthetic drugs have also been demonstrated to activate the protective function of Nrf2 in vascular ECs. Memantine promotes the activation of Nrf2/HO-1 antioxidant signaling pathway and protects cardiac ECs from dysregulation in acute myocardial infarction [83]. Metformin also increases the expression level of Nrf2 and the nuclear accumulation level of Nrf2 in hyperglycemic HUVECs. In addition, Metformin can downregulate p65 to upregulate Nrf2 as a way to protect ECs function associated with gestational diabetes mellitus (GDM) [84]. Other drugs that target Nrf2 and protect ECs are detailed in Table 1.

Although less-studied, a handful of evidence has revealed the abnormalities associated with Nrf2 activation in ECs. An early showed that prolonged infection of ECs with Kaposi’s sarcoma-associated herpesvirus (KSHV) induces Nrf2 activation, thereby promoting Kaposi’s sarcoma pathogenesis [85]. A recent study further documented that cigarette smoke-induced oxidative stress activates an Nrf2/STAT3 pathway to interrupt fibronectin assembly and angiogenesis in human umbilical vein ECs and zebrafish [86]. Unlike the Nrf2-mediated beneficial effects in ECs which are validated by Nrf2 EC-specific knockout approach [54], the ‘dark’ side of Nrf2 remains to be fully established by genetic interrogation in vivo. Further investigation of Nrf2-mediated dichotomy in ECs is warranted, therefore providing novel insight into the therapeutic implications of targeting Nrf2 signaling.

**Table 1 cells-11-03042-t001:** Therapeutic targeting Nrf2 signaling for EC protection.

Approach	Nrf2 Signaling	Finding	Reference
**Compounds**	**Atherosclerosis (AS) Related**		
Melatonin	Nrf2/NLRP3	Reduces cigarette smoke extract (CSE) treatment-induced pyroptosis in human aortic endothelial cells (HAECs) in vitro and cigarette smoke exposure-enhanced intimal hyperplasia in rat carotid arteries induced by balloon injury in vivo	[75]
Astragaloside	Nrf2/HO-1	Prevents oxidized low-density lipoprotein (oxLDL)-induced human umbilical vein endothelial cells (HUVECs) injury in vitro	[87]
Isoflavone	Nrf2/AER	Reduces oxLDL-induced oxidative stress damage in EA.hy926 cells in vitro and atherosclerosis in apolipoprotein E deficiency (ApoE^−/−^) mice fed with a high-fat diet in vivo	[88]
Flavonoids from a Deep-Sea-Derived *Arthrinium* sp.	AKT/Nrf2/HO-1	Protects HUVECs against ox-LDL-induced oxidative stress in vitro	[89]
Tanshinone IIA	Nrf2	Suppresses human coronary artery endothelial cells (HCAECs) ferroptosis in vitro	[90]
Chalcone derivative	Nrf2/HO-1	Inhibits cholesterol efflux and suppresses inflammatory responses in HUVECs in vitro Reduces lipid accumulation and plaque formation in LDL receptor knockout (Ldlr^−/−^) mice fed with a high-fat diet	[91]
Ilexgenin A	Nrf2/PSMB5	Suppresses mitochondrial fission, and improves endothelial dysfunction induced by palmitate (PA) in vitro	[92]
Acacetin	Nrf2	Exerts antioxidant potential in ApoE^−/−^ mice and in EA.hy926 cells induced by human oxLDL in vitro	[93]
Irisin	AKT/mTOR/S6K1/Nrf2	Attenuates oxLDL-impaired angiogenesis of human microvascular ECs in vitro and in a chicken embryo membrane (CAM) model in vivo	[94]
A peptide from microalgae *Isochrysis zhanjiangensis*	Nrf2	Inhibits oxLDL-induced inflammation and apoptosis of HUVECs in vitro	[95]
Kaempferol	PI3K/AKT/Nrf2	Attenuates oxLDL-induced injury via activating G protein-coupled estrogen receptor (GPER) associated with upregulation of PI3K/AKT/Nrf2 signaling in HAECs in vitro Suppresses atherosclerotic lesion formation in ovariectomized ApoE^−/−^ mice fed with a high-fat diet in vivo	[96]
Xanthoangelol	Nrf2/ARE	Prevents oxLDL-induced HUVECs injury in vitro	[97]
Equol	Nrf2/t-BHP/CHOP	Attenuates atherosclerosis in ApoE^−/−^ mice fed with a high-fat diet in vivo and endoplasmic reticulum stress and apoptosis in HUVECs induced by tert-butyl hydroperoxide (t-BHP) and thapsigargin in vitro	[98]
Kirenol	PI3K/AKT/Nrf2	Prevents B[a]P-induced redox imbalance in HUVECs in vitro	[99]
Resveratrol	Nrf2/ICAM-1	Suppressive effects on pro-inflammatory responses in ECs and accelerated atherosclerosis in carotid arteries induced by ApoE^−/−^ mice	[100]
Zedoarondiol	Nrf2/HO-1	Attenuates oxLDL-induced injury, oxidative stress, and inflammatory responses in HUVECs in vitro	[101]
Dietary ellagic acid	Nrf2/HO-1	Protective effects on damage in HAECs induced by hypochlorous acid (HOCl) in vitro and endothelial dysfunction in the mouse model of accelerated atherosclerosis in carotid arteries induced by partial ligation in vivo	[102]
Salvianolic acid B	Nrf2/HO-1	Inhibits tumor necrosis factor-alpha-induced NF-kappaB activation in HUVECs	[103]
Dihydromyricetin	ERK&AKT/Nrf2/HO-1	Protects HUVECs from oxLDL-induced oxidative injury in vitro	[104]
Miltirone	Nrf2/HO-1	Protects human EA.hy926 ECs from oxidative stress-associated injury induced by t-BHP and oxLDL in vitro	[105]
Z-Ligustilide	Nrf2/ARE	Protects EA.hy926 cells from t-BHP-induced oxidative stress in vitro and attenuates atherogenesis in Ldlr^−/−^ mice fed with a high-fat diet in vivo	[106]
	Nrf2/HO-1	Attenuates inflammatory responses in HUVECs in vitro	[107]
Vitexin	Wnt/beta-catenin and Nrf2	Protects HUVECs from high glucose-induced injury in vitro	[108]
Theaflavin	miR-24/Nrf2/HO-1	Alleviates oxidative injury and atherosclerosis progression in ApoE^−/−^ mice fed with a high-fat diet in vivo and protects against cholesterol-induced oxidative injuries in HUVECs in vitro	[109]
	PI3K/AKT/Nrf2	Attenuates t-BHP-induced oxidative stress in HUVECs in vitro and enhances vascularization in regenerated tissues and accelerates wound healing in vivo	[110]
β-Farrerol	GSK-3/Nrf2-ARE	Protect EA.hy926 cells against oxidative stress-induced injuries in vitro	[111]
**Compounds**	**Diabetes Related**		
Metformin	Nrf2	Ameliorates the inhibitory effect of high glucose on migration and angiogenesis of HUVECs in vitro	[84]
Carnosol	Nrf2/t-BHP	Protects against t-BHP-induced human retinal microvascular endothelial cells (HRMECs) injury	[112]
β-Buyang Huanwu Decoction	AKT/GSK3/Nrf2	Enhances revascularization in a mouse model of diabetic hindlimb ischemia (HLI)	[113]
Antrodin C	Nrf2/HO-1	Prevents hyperglycemia-induced senescence and apoptosis in human ECs in vitro	[114]
Allicin	Nrf2	Alleviates aortic inflammatory responses associated with type 1 diabetes induced by intraperitoneal injection of streptozotocin (STZ) in mice and high glucose-induced growth inhibition and death in HUVECs in vitro	[115]
**Compounds**	**Hypertension Related**		
Tilapia by-product oligopeptide	Nrf2/NF-κB	Protects against Ang II-induced hypertensive injury in HUVECs in vitro	[82]
A novel angiotensin-I-converting enzyme inhibitory peptide from microalgae *Isochrysis zhanjiangensis*	Nrf2	Inhibits Ang II-induced vascular factor secretion, inflammatory responses, and apoptosis in HUVECs in vitro	[116]
Hydrogen sulfide	Nrf2	Ameliorates endothelial dysfunction associated with hypertension in spontaneously hypertensive rats (SHR) and Ang II-induced cellular damage and ROS formation in HUVECs in vitro	[117]
**Compounds**	**Other Diseases Related**		
Celastrol	Nrf2/ERK1/2/Nox2	Attenuates Ang II-mediated HUVECs damage in vitro	[78]
Schisandrin C	Keap1/Nrf2	As an antioxidative agent for the treatment of Ang II-induced vascular endothelial deficits in vitro	[80]
Osthole	Keap1/Nrf2	Protects against Ang II-induced apoptosis of rat aortic endothelial cells (RAECs) in vitro	[81]
Memantine	Nrf2/HO-1	Protects against inflammatory responses and impaired endothelial tube formation induced by oxygen-glucose deprivation/reperfusion in HUVECs in vitro	[83]
Cyanidin-3-O-glucoside	Nrf2/Bach1 and NF-κB	Improves intracellular redox status of HUVECs exposed to palmitic acid (PA) in vitro	[118]
Panax notoginseng Saponin	PI3K/AKT/Nrf2	Protects against cerebral ischemia/reperfusion (I/R)-induced blood–brain barrier disruption in cerebral microvascular endothelial cells (bEnd.3) in vitro	[119]
Ginsenoside Rg3	Nrf2/ARE	Antagonizes adriamycin-induced cardiotoxicity by improving endothelial dysfunction both in vivo and in vitro	[120]
Procyanidin B2	Nrf2/PPARγ/sFlt-1	Ameliorates endothelial dysfunction and impaired angiogenesis of HUVECs in vitro and in a rat model of preeclampsia induced by uterine perfusion pressure (RUPP) in vivo	[121]
Rice bran phenolic Compounds	Nrf2/HO-1/NQO1/eNOS	An antioxidant/anti-inflammatory effect on HUVECs with induced oxidative stress in vitro	[122]
Chlorogenic acid	Nrf2/HO-1	Protective effects on Ang II infusion-induced vascular senescence in mice and H_2_O_2_-induced senescence in HUVECs in vitro	[123]
Pterostilbene	Keap1/Nrf2/HO-1	Protects against uraemic serum (US)-mediated injury in HUVECs in vitro	[124]
Lipoxin A4	Nrf2/HO-1	Attenuates H_2_O_2_-evoked cytotoxic injury in HUVECs in vitro	[125]
Hydrogen sulfide	Nrf2/HIF-1α	Prevents balloon injury-induced neointimal hyperplasia in carotid arteries of rats in vivo and enhances HUVECs tube formation and migration in vitro	[126]
Paeoniflorin	Nrf2/HO-1	Alleviates t-BHP-stimulated HUVECs cellular dysfunction and apoptosis in vitro and enhances the vascularization of regenerated tissues and promotes flap survival in vivo	[127]
Irigenin	Nrf2	Alleviates Ang II-induced oxidative stress and apoptosis in HUVECs in vitro	[128]
Cinnamaldehyde	Nrf2/HO-1	Protects against H_2_O_2_ or TNFα-induced inflammatory responses in HUVECs and inhibits lipopolysaccharide (LPS)-induced inflammatory cell infiltration in vivo	[129]
Protandim	Nrf2	Protects against salt-induced vascular dysfunction in vivo by restoring redox homeostasis in the vasculature	[130]
Magnesium lithospermate B	PI3K/AKT/Nrf2	Protects against LPS-induced endothelial dysfunction in a murine acute inflammation model and in human dermal microvascular endothelial cells (HMECs-1) in vitro	[131]
Ginkgo biloba extract	Nrf2/HO-1	Reduces leukocyte adherence to injury arteries, enhances HO-1 expression in circulating monocytes and arteries after wire injury, and reduces TNF-alpha-stimulated endothelial adhesiveness	[132]
Brazilian Green Propolis	PI3K/AKT/mTOR/Nrf2/HO-1	Inhibits oxLDL-stimulated oxidative stress in HUVECs in vitro	[133]
Unripe Carica papaya fruit extract	Nrf2	Protects ECs challenged with H_2_O_2_ in vitro	[134]
Lycopene	Nrf2/HO-1	Inhibits cyclic strain-induced ET-1 gene expression of HUVECs in vitro	[135]
Hippocampus abdominalis-derived peptides	Nrf2/HO-1	Protects H_2_O_2_-induced cell death through antiapoptotic action in vitro	[136]
Aspirin eugenol ester	Nrf2	Attenuates oxidative injury in a hamster model of atherosclerosis induced by a high-fat diet and H_2_O_2_-induced apoptosis in HUVECs, an in vitro model of oxidative stress	[137]
**miRNAs**			
miR-24	Nrf2/HO-1	Promotes endothelial repair caused by oxidative stress after balloon injury in diabetic rats	[138]
miR-200a	Nrf2	Thymosin beta-4 (Tbeta4) attenuates H/R-induced cardiac microvascular endothelial cells (CMECs) injury by miR-200a-Nrf2 signaling in vitro	[139]
miR-140-5p	Nrf2 and Sirt2	Upregulation of miR-140-5p in the atherosclerotic aorta in ApoE^−/−^ mice Increases oxidative stress in HUVECs in vitro	[140]

### 3.2. Nrf2 and Vascular Smooth Muscle Cells

Vascular smooth muscle cells (VSMCs) are not terminally differentiated, instead, they are highly plastic. Under pathological conditions, mature quiescent and contractile VSMCs can be converted into a synthetic type of VSMCs characterized by decreased expression of SMC contractile proteins and increased capabilities of cellular proliferation, migration, and synthesis and secretion of extracellular matrix, calcific cells, or even stem cells, a process known as VSMC phenotypic transition or dedifferentiation which contributes to the development and progression of many vascular diseases [141,142,143,144,145].

Single-cell RNA sequencing shows that Nrf2 is a key regulator of VSMC dedifferentiation [141]. Nrf2 is required for choline-induced inhibition of VSMC dedifferentiation and vascular protection [146]. Other evidence reveals that downregulation of Nrf2 expression and activity contributes to the dedifferentiation of coronary arterial smooth muscle cells due to the genetic deficiency of CD38 [147]. Whilst Nrf2 depletion enhances platelet-derived growth factor (PDGF)-stimulated migration but not the proliferation of rat aortic SMCs, global Nrf2 knockout inhibits intimal hyperplasia of wire-injured femoral arteries in mice [148]. Accordingly, Nrf2 signaling becomes the target of botanical medicine for the treatment of vascular disease (Table 2) [149,150,151,152,153,154,155]. For example, the Keap1-Nrf2-ARE antioxidant system was demonstrated to mediate the inhibitory effects of rosmarinic acid, a hydroxylated compound frequently found in herbal plants, on platelet aggregation, VSMC dedifferentiation, and neointima formation [149]. Nrf2 activation is associated with andrographolide-induced suppression of pulmonary arterial SMC growth in vitro and pulmonary arterial pathological remodeling and hypertension in vivo [150]. Similarly, cinnamic aldehyde, an electrophilic Nrf2 activator, can inhibit VSMC growth and intimal hyperplasia after balloon injury in a rat model of diabetic restenosis [152]. In addition, such beneficial activation of Nrf2 is implicated in the potential efficacy of other plant extracts in treating vascular diseases, including prunella vulgaris, sulfiredoxin-1, and physalin B [103,151,156,157,158,159,160]. Synthetic drugs have also been documented to activate the protective function of Nrf2 in VSMCs. For example, coenzyme Q10 (CoQ10), a commonly used nutritional supplement, promotes the expression of Nrf2 and reduces oxidative stress in VSMCs, and attenuates intracranial aneurysm formation and rupture in mice [161]. Canagliflozin, a new sodium-glucose co-transport protein 2 (SGLT2) inhibitor, reduces CVD and mortality in patients with type 2 diabetes presumably via the activation of ROS/Nrf2/HO-1 path way in VSMCs [162]. An axis of ERK5/Nrf2 activation induced by statins such as fluvastatin reduces advanced glycation endproduct-induced VSMCs proliferation and migration, thus supporting the therapeutic potential of targeting the ERK5-Nrf2 signaling module in treating vascular lesions associated with diabetes [163]. Other functional synthetic drugs that protect VSMCs by activating Nrf2 are listed in Table 2.

Vascular calcification, a common complication of CVD, is driven in part by VSMC dedifferentiation into calcific cells [170]. Interestingly, several studies demonstrated that Nrf2-operated antioxidant defense [171,172] is critical for suppressing VSMC dedifferentiation into calcific cells in vitro. In addition, the gastransmitter hydrogen sulfide (H_2_S) attenuates VSMC calcification through activation of Keap1/Nrf2/NQO1 [164] while the rosmarinic acid- or mitoquinone-induced suppression of VSMC calcification depends on Nrf2 activation [166,168]. In contrast, O-linked N-acetylglucosamine transferase (OGT)-mediated Keap1 glycosylation accelerates the degradation of Nrf2 and suppresses autophagy. thereby leading to hyperphosphatemia-induced vascular calcification in chronic kidney disease [173].

However, emerging evidence indicated that activation of Nrf2 likely underlies palmitate-induced pro-inflammatory responses in human coronary artery SMCs in vitro [174]. In addition, Nrf2 deficiency can attenuate atherosclerosis by reducing lectin-like oxidized low-density lipoprotein receptor (LOX)-1-mediated VSMC proliferation and migration [143]. Whether such Nrf2-operated pathological signaling is restricted to VSMCs in a proatherosclerotic setting remains unknown.

Clearly, activation of Nrf2 activation can be either beneficial or detrimental to VSMCs, thereby contributing to either vascular repair or disease. Further studies of molecular mechanisms driving the Nrf2-operated dichotomy in diverse pathophysiological settings may shed light on the complexity of Nrf2 signaling in the vasculature.

### 3.3. Nrf2 and Vascular Disease

Atherosclerosis is a chronic vascular disease of the arteries. Oxidative stress is an important factor in the development of atherosclerosis, and the Nrf2 signaling pathway as the main antioxidant pathway has become an important target for atherosclerosis prevention and treatment [175]. Intriguingly, it has been demonstrated that Nrf2 plays a dual role in the development and progression of atherosclerosis (Table 3). Nrf2 knockout approach revealed that Nrf2 upregulates the expression of receptor CD36 and enhances uptake of oxLDL and foam cell formation in macrophages, thereby promoting atherosclerosis development [176,177,178]. Nrf2 exacerbated atherosclerosis by enhancing IL-1-mediated vascular inflammation [179]. In addition, Nrf2 participated in the development and progression of atherosclerosis by promoting the polarization of macrophages [180,181]. Of note, these Nrf2-mediated detrimental actions occur in the setting of ApoE deficiency. However, in hypercholesterolemic mice, Nrf2 can mediate both pro- and anti-atherosclerotic effects. Although the Nrf2-CD36 pathway promotes atherosclerosis, Nrf2-mediated induction of antioxidant genes may contribute to a reduction in atherosclerotic lesion development [182,183].

Abdominal aortic aneurysm (AAA) is a degenerative disease that is one of the principal causes of death in people over 65 years of age. The increasing ROS and oxidative stress seem to play a key role in the development of AAA [184]. Many drugs are designed to treat AAA by targeting Nrf2-operated antioxidant defense [185,186]. For example, calcitriol supplementation reduces the severity of AAA by reactivating Nrf2 and inhibiting apoptotic pathways. Additionally, betanin prevents experimental AAA progression by modulating the Nrf2/HO-1 pathways. Several small molecules and metabolites also protect AAA progression by targeting Nrf2. For example, itaconate prevents abdominal aortic aneurysm formation by inhibiting inflammation through the activation of Nrf2 [187]. Sestrin2 attenuates Ang II-induced apoptosis in VSMCs via the Nrf2 pathway of AAA [188]. Carbon monoxide-induced Nrf2/HO-1 alleviates inflammatory responses to Ang II by inhibiting NADPH oxidase- and mitochondria-derived ROS in AAA [189]. However, simvastatin treatment increases HO-1 protein levels in AAA, but independently of Nrf2 [190]. When the Nrf2 transcriptional activity is lacking, simvastatin may further reduce AAA formation [191].

**Table 3 cells-11-03042-t003:** Nrf2 signaling cascades in vascular damage and dysfunction.

Nrf2 Signaling	Animal Model	Pathological Setting	Treatment	Phenotype	Proposed Mechanism	Reference
Nrf2/CD36	Nrf2^−/−^, ApoE^−/−^	Atherosclerosis	None	Nrf2 knockout decreases susceptibility to atherosclerotic plaque formation in ApoE^−/−^ mice.	Nrf2 upregulates CD36 thereby promoting foam cell formation and the pathogenesis of atherosclerosis.	[177]
Nrf2/CD36	Nrf2^−/−^, ApoE^−/−^	Atherosclerosis	None	Nrf2 expression promotes atherosclerotic lesion formation.	Nrf2 deficiency results in decreased cholesterol Influx, correlated with lower CD36 expression.	[178]
Nrf2/IL-1	Nrf2^−/−^, ApoE^−/−^	Atherosclerosis	None	Nrf2-deficient ApoE^−/−^ mice are highly protected against diet-induced atherogenesis.	Nrf2 aggravates atherosclerosis by enhancing IL-1-mediated vascular inflammation.	[179]
Nrf2	Nrf2^−/−^, ApoE^−/−^	Atherosclerosis	None	Nrf2 deletion in bone marrow-derived cells is protective against atherosclerosis.	Nrf2^−/−^; ApoE^−/−^ decreases expression of macrophage M1-subtype genes in mice.	[180]
Nrf2	ApoE^−/−^	Atherosclerosis	PCB29-pQ	PCB29-pQ driven CD163+ macrophage accumulated in the aortic valve.	Nrf2 activation is the main reason for macrophage differentiation.	[181]
Nrf2		Atherosclerosis	Vitamin E	Nrf2 can mediate both pro-and anti-atherosclerotic effects.	Nrf2 activation can promote CD36-mediated cholesterol uptake by macrophages, increase induction of Nrf2-mediated antioxidant genes, and is likely to contribute to decreased lesion progression.	[182]
Nrf2	Nrf2^−/−^, Ldlr^−/−^, Apob^100/100^	Atherosclerosis	None	Nrf2 deficiency impairs atherosclerotic lesion development but promotes the features of plaque instability in hyper-cholesterolemic mice.	Nrf2 deficiency in Ldlr^−/−^ mice reduces total plasma cholesterol and triglycerides in Ldlr^−/−^ApoB^100/100^ mice and aggravates aortic plaque maturation as it increased plaque calcification.	[183]
Nrf2	Nrf2^−/−^	Abdominal aortic aneurysm	None	Lacking Nrf2 transcriptional activity attenuates AAA formation under simvastatin treatment.	The inhibition of Nrf2 transcriptional activity facilitates AAA formation in mice, which can be prevented by simvastatin.	[191]

## 4. Therapeutic Potential of Targeting Nrf2 Pathway

Nrf2 can be a potential therapeutic target for a variety of diseases, including CVDs, thus the development of drugs that modulate Nrf2 activity or pathways of actions appears to be an important approach for the future clinical treatment of CVDs. This notion is indeed supported by a mounting number of previous studies showing Nrf2-mediated cardiovascular protection in diverse pathological settings. However, as we recently highlighted in the heart [14], Nrf2 activation may not always be beneficial in the vasculature aforementioned. Of note, in ApoE^−/−^ mice, knockdown of Nrf2 has a protective effect against atherosclerosis [177,178,179,180]. While it is generally accepted that Nrf2-mediated vascular protection is due to Nrf2-operated antioxidative defense, it remains unclear how Nrf2 exacerbates vascular disease. A series of our studies demonstrated that in the heart, Nrf2 functions protectively when autophagy is normal, but when autophagy is impaired, the Nrf2-mediated cardiac protection is lost, and Nrf2-operated myocardial damage appears [11,12,13]. Therefore, it is most likely that inhibition of autophagy is responsible for the activation of Nrf2-mediated cardiac injury. Of interest, several studies documented that autophagy is inhibited in atherosclerosis [192] and autophagy activation suppresses atherosclerosis in ApoE^−/−^ mice [193,194]. Accordingly, it is possible that autophagy also plays a critical role in determining the biological consequences of Nrf2 activation in the vasculature. Taken together, these findings reveal Nrf2-mediated dichotomy in the cardiovascular system. Special precautions should be taken to avoid the detrimental activation of Nrf2.

Notably, since methyl Bardoxolone, a potent Nrf2 activator, increases the incidence of cardiovascular events, including heart failure and death, the clinical phase III trial testing its efficacy in the treatment of chronic kidney disease associated with type 2 diabetes was terminated [195]. Although the precise reasons remain to be determined, a possibility is that Bardoxolone methyl activates a yet unrecognized Nrf2 detrimental signaling in such a pathological setting. Unfortunately, the pathological activation of Nrf2 is still far from a comprehensive understanding. Whilst it is important to study the endogenous signaling pathways underlying Nrf2-mediated cardiovascular protection such as Notch-Nrf2 axis and PI3K-AKT-Nrf2 pathway in the cardiovascular system [50,196], it is crucial to dissect the Nrf2 signaling that causes cardiovascular damage and dysfunction. Accordingly, the outcome could provide guidance for the rationale, design, and screen of a novel class of Nrf2 modulators that can selectively activate Nrf2-mediated cardiovascular protection while turning off the Nrf2-operating pathological program.

## 5. Closing Remark

Since the discovery of Nrf2 as a master transcription factor of antioxidant defense, the Nrf2 signaling pathway has become not only a hot topic in the field of anti-oxidative stress research but also a therapeutic target to treat various diseases. It is clear that Nrf2 is a multifunctional transcription factor and executes diverse biological actions in a context-dependent manner, resulting in either protective or detrimental effects on the cardiovascular system. However, the molecular mechanisms which turn on or off the Nrf2-operating dichotomous actions are poorly understood. Nevertheless, there are still a lot of efforts in developing Nrf2 activators to treat several non-CVDs [197]. Therefore, it is necessary to further investigate the molecular mechanisms of Nrf2-mediated cardiovascular damage and dysfunction, thereby providing crucial information for precisely targeting Nrf2 signaling to treat cardiovascular and other diseases.

## Figures and Tables

**Figure 1 cells-11-03042-f001:**
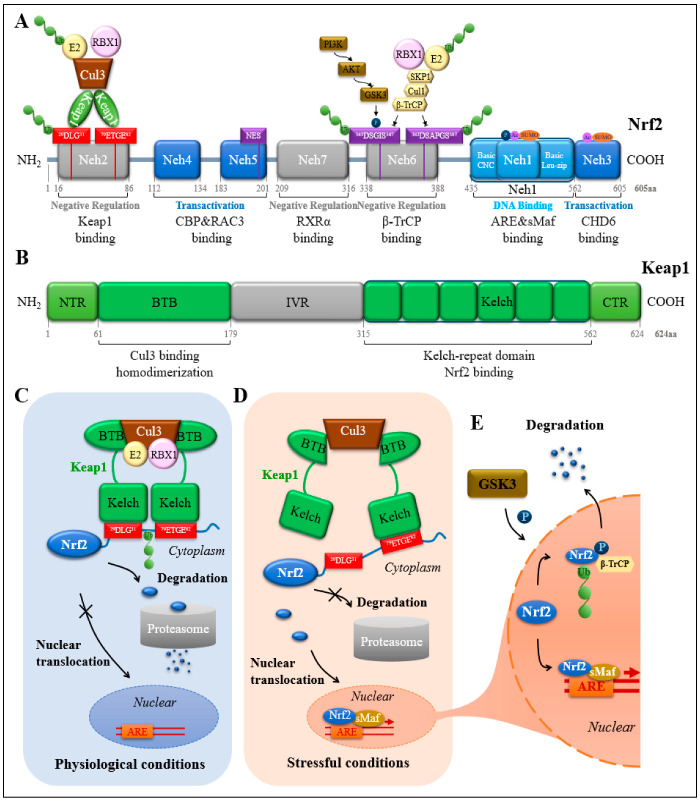
Detailed schematic diagram of Nrf2-related signaling pathway. Structural characteristics of Nrf2 (**A**) and Keap1 (**B**). The spatial patterns of interaction between Nrf2 and Keap1 under physiological conditions (**C**) and stressful conditions (**D**). The degradation of Nrf2 depends on different proteins under different conditions. Under physiological conditions, it depends on Keap1 (**C**); under stressful conditions, it depends on GSK-3 (**E**). Ub, ubiquitin; P, Phosphorylation; SUMO, small ubiquitin-like modifiers, SUMOylation; Ac, acetylation.

**Table 2 cells-11-03042-t002:** Therapeutic targeting Nrf2 signaling for VSMC protection.

Approach	Nrf2 Signaling	Finding	Reference
**Compounds**	**Atherosclerosis (AS) Related**		
Salvianolic acid B	Nrf2/HO-1	Inhibits PDGF-induced proliferation and migration of VSMCs Activates Nrf2-mediated HO-1 expression and suppression of PDGF-induced neointimal hyperplasia in cultured arterial rings isolated from mice ex vivo	[103]
Eupatolide	Nrf2/HO-1	Inhibits PDGF-induced proliferation and migration of aortic SMCs in vitro	[159]
**Compounds**	**Diabetes Related**		
Aqueous extract of *Prunella vulgaris*	Nrf2/HO-1	Exhibits inhibitory effects on high glucose-stimulated VSMCs proliferation and migration, and invasion activities in vitro	[151]
Cinnamic aldehyde	Nrf2	Inhibits neointimal hyperplasia after carotid artery balloon injury in the Zucker Diabetic Fatty (ZDF) rats and inhibits proliferation of ZDF VSMCs in vitro	[152]
Canagliflozin	Nrf2/HO-1	Stimulates HO-1 expression in mice and human VSMCs through the ROS-Nrf2 pathway, and inhibits VSMCs proliferation and migration in vitro	[162]
Fluvastatin	ERK5/Nrf2	Activates ERK5-dependent Nrf2 pathway and inhibits cellular proliferation and migration in VSMCs in vitro	[163]
**Compounds**	**Vascular Calcification Related**		
Hydrogen sulfide	Keap1/Nrf2/NQO1	Attenuates circulating calciprotein particles (CPP)-induced VSMCs calcification in vitro	[164]
Metformin	Nrf2	Inhibits hyperlipidemia-associated calcium deposition in the rat aortic tissue of hyperlipidemia-related vascular calcification model in vivo and attenuates ferroptosis with increased calcium deposition in VSMCs with PA treatment	[165]
Mitoquinone	Keap1/Nrf2	Attenuates vascular calcification by suppressing oxidative stress and reducing apoptosis in adenine-induced calcification in rats and inorganic phosphate-induced calcification in VSMCs in vitro	[166]
miR-126	Sirt1/Nrf2	Attenuates calcification, in human aortic smooth muscle cells (HASMCs) in vitro and in a mouse calcification model in vivo	[167]
Rosmarinic acid	Keap1/Nrf2/ARE	Inhibits VSMCs proliferation, migration, and calcification in a rat model of vascular calcification model induced by high-fat diet and vitamin D3 injection as well as β-glyerophosphate-induced calcification in rat aortic SMCs in vitro	[168]
**Compounds**	**Vascular Occlusive Disease and Other Diseases Related**		
Rosmarinic acid	Keap1/Nrf2/ARE	Inhibits platelet aggregation and neointimal hyperplasia in vivo and VSMCs dedifferentiation, proliferation, and migration in vitro	[149]
Andrographolide	NOX/Nrf2	Reverses pulmonary vascular remodeling through modulation of NOX/Nrf2-mediated oxidative stress and NF-κB-mediated inflammation in both chronic hypoxia and Sugen5416/hypoxia mouse pulmonary hypertension (PH) models and in cultured human PASMCs isolated from either healthy donors or PH patients	[150]
Sulforaphane	NOX4/ROS/Nrf2	Attenuates Ang II-induced human VSMCs migration in vitro	[153]
Trans-resveratrol	Nrf2/HO-1	Suppresses intimal hyperplasia in a mouse model of wire-injured femoral artery injury by oral administration Inhibits PDGF-stimulated DNA synthesis and cell proliferation in VSMCs in vitro	[155]
Sulfiredoxin-1	Nrf2/ARE	Inhibits PDGF-BB-induced VSMCs proliferation and migration in vitro	[156]
Physalin B	Nrf2/HO-1	Inhibits PDGF-BB-induced VSMCs proliferation, migration, and phenotypic transformation in vitro and prevents intimal hyperplasia in a mouse model of carotid artery injury induced by ligation	[157]
Gemigliptin	Nrf2/HO-1	Exerts a preventative effect on ligation injury-induced neointimal hyperplasia in vivo and inhibits VSMCs proliferation and migration in vitro	[158]
Malabaricone C	Nrf2/HO-1	Inhibits PDGF-induced proliferation and migration of aortic SMCs in vitro	[160]
Coenzyme Q10	Nrf2/HO-1	Attenuates intracranial aneurysm formation and rupture in mice and reduces H_2_O_2_-induced oxidative stress in VSMCs in vitro	[161]
Sulfasalazine	Nrf2/HO-1	Suppresses VSMCs growth in vitro and prevents neointimal hyperplasia in rat carotid arteries induced by balloon in vivo	[169]

## Data Availability

Not applicable.

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
