# Peer review of "Nrf2-Mediated Dichotomy in the Vascular System: Mechanistic and Therapeutic Perspective"

_cells, 2022, doi:10.3390/cells11193042_

Round 1

Reviewer 1 Report

This review described the ubiquitous nature of Nrf-2, which has been thought to play an important role in various diseases in recent years, and its association with cardiovascular disease. The viewpoints selected included many areas that were still unknown in previous researches. In addition, the contents presented covered new findings and was highly suggestive for future research. However, there are some points which the authors should addressed as follows;

1.     There are some typographical and grammatical errors in the manuscript. The authors should correct them throughout.

2.     Please explain in more detail why the authors thought that Nrf2 was deeply involved in cardiovascular disorders.

3.     Part of the legend in Figure 1 was obscured. Also, please simplify the legend because it seems a little too complicated.

4.     Please make a figure of the Keep1-Nrf2 cascade (pathway) for better understanding.

5.     Was there only one reference paper to cite in Section 4?

Reviewer 2 Report

In this paper, the authors have reviewed the role of Nrf2 as therapeutic target in cardiovascular diseases, dealing with not only its beneficial but also its detrimental effects. This dual role of Nrf2 is a very  hot topic, not only in cardiovascular field, and deserves due attention by scientists. In this view, the paper by Wu et al. is an interesting work.

The paper is clearly written and well structured: the authors first describe the biomolecular and functional characteristics of Nrf2, then they report the positive and negative effects of Nrf2 activation on the cytological components of blood vessels, and finally its role in vascular and cardiac diseases.

I have just a few remarks.

Major:

1)      The mechanisms thought to be responsible for both positive and negative effects of Nrf2 activation have been extensively explained for atherosclerosis, but scarce information has been given for other cases, especially about the detrimental effects. Although the underlying mechanisms are often unclear, more details should be given for e.g.  cigarette-smoke effects in vascular formation or pro-inflammatory response in human coronary artery smooth muscle cells.

2)      In Table I the effects of many drugs are reported, but in the text only the effects of natural compounds are described. The authors should add to the text some examples of synthetic drugs. Moreover, in Table II only botanical compounds are cited. Why? Examples of Nrf2 activation by synthetic drugs should be added, if any.

3)      The chapter “3.4. Nrf2 and Cardiac Disease” is quite poor in comparison to the other ones, the authors should add more details.

Minor:

1)      The tables should be reorganized to improve their readability. For instance, where possible, the lines describing the same disease or Nrf2 signaling could be combined.

2)      The legend of figure 1 is not fully readable.

3)      Cardiovascular disease has been abbreviated as CVD at the beginning of the text, but the abbreviation has not been used consistently through the text.

4)      Some sentences are too similar to previous published works and should be rephrased to avoid plagiarism or duplication.

5)      There are several typos through the text, please check.
